# Trametinib Induces the Stabilization of a Dual *GNAQ* p.Gly48Leu- and *FGFR4* p.Cys172Gly-Mutated Uveal Melanoma. The Role of Molecular Modelling in Personalized Oncology

**DOI:** 10.3390/ijms21218021

**Published:** 2020-10-28

**Authors:** Fanny S. Krebs, Camille Gérard, Alexandre Wicky, Veronica Aedo-Lopez, Edoardo Missiaglia, Bettina Bisig, Mounir Trimech, Olivier Michielin, Krisztian Homicsko, Vincent Zoete

**Affiliations:** 1Computer-aided molecular engineering group, Department of Fundamental Oncology, Lausanne University, Ludwig Lausanne Branch, 1066 Epalinges, Switzerland; fanny.krebs@unil.ch; 2Precision Oncology Center, Department of Oncology, Lausanne University Hospital, 1011 Lausanne, Switzerland; camille.gerard@chuv.ch (C.G.); alexandre.wicky@chuv.ch (A.W.); olivier.michielin@chuv.ch (O.M.); krisztian.homicsko@chuv.ch (K.H.); 3Service of Medical Oncology, Department of Oncology, Lausanne University Hospital, 1011 Lausanne, Switzerland; veronica.aedo-lopez@chuv.ch; 4SIB Swiss Institute of Bioinformatics, 1015 Lausanne, Switzerland; edoardo.missiaglia@chuv.ch; 5University Institute of Pathology, Lausanne University Hospital, 1011 Lausanne, Switzerland; bettina.bisig@chuv.ch (B.B.); mounir.trimech@chuv.ch (M.T.); 6Laboratory of Translational Oncology, EPFL, 1015 Lausanne, Switzerland

**Keywords:** precision oncology, molecular modelling, mutation, GNAQ, FGFR4

## Abstract

We report a case of an uveal melanoma patient with *GNAQ* p.Gly48Leu who responded to MEK inhibition. At the time of the molecular analysis, the pathogenicity of the mutation was unknown. A tridimensional structural analysis showed that Gα_q_ can adopt active and inactive conformations that lead to substantial changes, involving three important switch regions. Our molecular modelling study predicted that *GNAQ* p.Gly48Leu introduces new favorable interactions in its active conformation, whereas little or no impact is expected in its inactive form. This strongly suggests that *GNAQ* p.Gly48Leu is a possible tumor-activating driver mutation, consequently triggering the MEK pathway. In addition, we also found an *FGFR4* p.Cys172Gly mutation, which was predicted by molecular modelling analysis to lead to a gain of function by impacting the Ig-like domain 2 folding, which is involved in FGF binding and increases the stability of the homodimer. Based on these analyses, the patient received the MEK inhibitor trametinib with a lasting clinical benefit. This work highlights the importance of molecular modelling for personalized oncology.

## 1. Introduction

*GNAQ* codes for the Gαq cytoplasmic protein, which belongs to the G protein-coupled receptor family (GPCR) [1]. In its inactive conformation, Gαq binds Guanosine 5′-diphosphate (GDP) and associates with the two subunits Gβ and Gγ (Appendix A), forming the heterotrimeric G protein, which binds to the cytoplasmic GPCR transmembrane loops. An external stimulation of GPCR induces conformational changes in the GPCR transmembrane segments that are transmitted to the bound heterotrimeric G protein, ultimately promoting the release of GDP in favor of guanosine 5′-triphosphate (GTP) binding [2]. This activates Gαq which can regulate cellular machinery [3]. Most pathologic mutants involve hotspots of Arg183 and Gln209 in Gαq switch I and II regions, respectively. Both participate in GDP/GTP interactions (Figure 1a,b). *GNAQ* or *GNA11* mutations are found in >90% of uveal melanomas, mutually exclusive of one another, most commonly affecting the Gln209 hotspot. Other recurrently mutated genes in this tumor type are *BAP1*, *SF3B1* and *EIF1AX* [4,5,6] (Appendix A).

*FGFR4* encodes the protein fibroblast growth factor receptor which is one of the four highly conserved transmembrane members of the fibroblast growth factor receptor family (FGFRs) (Appendix A) The protein structure consists of an extracellular domain (ECD) composed of three immunoglobulin-like (Ig-like) domains (D1, D2 and D3), followed by a transmembrane helix and a cytoplasmic tyrosine kinase domain. The native ligands of FGFRs are fibroblast growth factors (FGFs) that bind to D2 and D3 in the ECD in the presence of heparin or heparin sulfate cofactors. It has not been clearly demonstrated if this binding induces the receptor dimerization or if the ligands bind to the receptor homodimer. This provokes an important conformation change that triggers the intracellular kinase domain autophosphorylation, which leads to the activation of MAPK, PLCγ and STATs pathways [9]. FGFR members are quite often implicated in various types of cancers and *FGFR4* mutations are present in 6% of melanoma [6].

*GNAQ* c.142_143delinsTT (p. Gly48Leu) and *FGFR4* c.514T > G (p.Cys172Gly) were detected by next-generation sequencing (NGS) in a subcutaneous metastasis of a patient known to have a uveal melanoma, and presented in the weekly molecular tumor board (MTB) at Lausanne University Hospital (CHUV), Switzerland. Since no information existed in the publicly available literature on their functional significance, the possible impact of these alterations on the respective protein structure and activity was analyzed by molecular modelling, which predicted a probable activation of both proteins, supporting a treatment based on MEK inhibitors [10,11,12]. Thus, a targeted therapy with trametinib was proposed, which triggered a partial positive response of the patient, stabilizing the cancer. This result, which constitutes a striking example of a computer-to-bed application of molecular modelling, supports its role in oncology-related MTBs.

## 2. Results

### 2.1. Modelling Analysis

#### 2.1.1. *GNAQ* p.Gly48Leu

Gαq’s 3D structure is composed of an α-helical domain and a RAS-like domain which houses GTPase activity and is required for the Gβγ heterodimer binding (Figure 1a). Three switch regions (SW-I, SW-II, SW-III) in the RAS-like domain play crucial roles in the protein activity (Figure 1). Interestingly, *GNAQ* p.Gly48 corresponds to *H/K/NRAS* p.Gly12, a famous mutation hotspot in RAS (Appendix A) [13]. In the inactive conformation of Gαq, SW-I interacts with GDP, while SW-II interacts with the Gβγ heterodimer. In the active conformation of Gαq, these switches adopt appropriate positions to bind GTP, leading to a more compact structure (Figure 1c and Appendix A).

Three mutations of Gly48 have been previously identified [6,14,15,16]. p.Gly48Val was found in cherry hemangiomas in the presence of a hotspot mutation in position Gln209, and no effect was attributed to it [17]. The second, which was published after our modelling analysis, is the p.Gly48Leu mutation itself. It was found in three cases of a hepatic small vessel neoplasm in the absence of other detected pathogenic or likely pathogenic mutations and was therefore suggested to have a potentially activating role, although it was not characterized [18]. The third one is the nonsense mutation Gly48*, which means that no other residue is expressed after position 48, leading to a loss of function as essential domains are deleted.

Gly48 is a highly conserved residue in many organisms, such as the switch’s regions (Appendix A). Six experimental structures, Gαq from Mus Musculus, are present in the Protein Data Bank, all of them in an active conformation. Three other 3D structures contain a guanine nucleotide-binding protein G(i) subunit alpha-1/guanine nucleotide-binding protein G(q) subunit alpha chimeric protein (Gαi/Gαq) from *Mus Musculus x Rattus Norvegicus*. Gαi and Gαq share very similar sequences and functions. Their overall structural conformations are identical and both catalyze the exchange of GDP to GTP. The sequence identity between these domains and human Gαq ranges from 94.1 to 99.7% (Appendix A). Furthermore, from the alignment of the structure sequences with human *GNAQ*, we can see that even the Gαi/Gαq chimera are highly similar to human sequence as the only alignment difference is before position 25 based on the human sequence and it does not affect the active site structure, nor the switch region structures. The totality of the binding site and switch regions are resolved in all structures used for this study. Structural analysis showed that Gly48 is situated in the active site of Gαq, close to SW-II. In the inactive conformation, Gly48 is surrounded within 5 Å by Gly46, Thr47, Glu49, Ser50, Lys52, the GDP phosphate tail, Glu234 and Arg247 (Figure 2a and Appendix A). In the active conformation, the same residues are found in the vicinity of Gly48, together with other residues from SW-II: Gly208, Gln209 and Arg213 (Figure 2b and Appendix A). The environment of Gly48 in the active conformation is more compact and includes SW-II. Due to time limitations in the context of a molecular tumor board, time-consuming modelling methods such as molecular dynamics simulations could not be used to analyse the impact of the mutation on the protein. Hence, we estimated the impact of the p.Gly48Leu mutation on the folding free energy of the Gαq and Gαi/Gαq structures with the FoldX software [19], a well-known and efficient program for predicting changes in free energy of folding upon mutations [20], whose predictive efficiency has been trained on a large set of mutants covering most of the existing structural environment. This calculation was repeated for all structures, in the active and inactive conformations, resulting in the folding free energy change upon mutation of both forms of the proteins. Calculated values are reported in Scheme 1 and Appendix A.

These results indicate that mutation p.Gly48Leu is favorable to the structural stability of Gαq both in the active and inactive conformations. However, the effect is more pronounced in the active form (1.8 ± 0.7 kcal/mol) than in the inactive one (0.4 kcal/mol). Analyses of the mutant structures generated by FoldX suggest that p.Gly48Leu active conformation is surrounded by the same residues as Gly48, plus Glu34 and Arg247 as well as Leu239, Val240 and Glu241 from SW-III. For each active conformation, the best conformer of the p.Gly48Leu mutant is oriented toward the SW-III region, which allows hydrophobic interactions with Leu239 and Val240 and thus contributes to stabilizing SW-III close to the active site, which favors the active conformation. This interaction does not exist in the inactive conformation of the mutant, where SW-III is too far from Leu48. In the active conformation, the median distances between Leu48 and Leu239, or Leu48 and Leu249 are 6.3 and 5.3 Å, respectively (Figure 3, Appendix A). These median distances are 15.8 and 8.8 Å, respectively, in the inactive structure. All these findings argue in favor of a possible activation of Gαq as a result of the *GNAQ* p.Gly48Leu mutation.

#### 2.1.2. *FGFR4* p.Cys172Gly

FGFR4′s structure consists of an extracellular domain (ECD) composed of three immunoglobulin-like (Ig-like) domains (D1, D2 and D3), followed by a transmembrane helix and a cytoplasmic tyrosine kinase domain. The native ligands of FGFRs are fibroblast growth factors (FGFs) that bind to D2 and D3 in the ECD in the presence of heparin or heparin sulfate cofactors. It has not been clearly demonstrated if this binding induces the receptor dimerization or if the ligands bind to the receptor homodimer. This leads to an important conformation change that triggers the intracellular kinase domain autophosphorylation, leading to the activation of the MAPK, PLCγ and STATs pathways [9].

Structural analysis shows that *FGFR4* p.Cys172 is buried in Ig-like D2 in the ECD, and participates in a disulfide bridge with Cys224 from the same domain. The latter is essential for the structural stability of D2, which plays a role in FGF binding [21] (Figure 4 and Appendix A). Due to glycine being unable to reproduce Cys172 interactions, mutation *FGFR4* p.Cys172Gly is predicted to have a severe structural impact on Ig-like D2. Multiple sequence alignments of human FGFR4 with orthologs show that this position is highly conserved (Appendix A). *FGFR4* p.Cys172 is also conserved in the whole FGFR family and corresponds to Cys178, -179 and -176 in *FGFR1*-2 and -3, respectively (Figure 4c). *FGFR1* p.Cys178Ser is already reported in the literature and predicted to lead to a gain of function as it demonstrates a constitutive activation of the FGFR1 dimer state in vitro [22]. Indeed, it was inferred that this mutation stabilizes the homodimer, favouring the active conformation of the transmembrane dimers. There is no mutation reported for *FGFR4* p.Cys179 and *FGFR3* p.Cys176 positions. However, the recently reported mutation *FGFR2* p.His167_Asn173del demonstrated oncogenic transformation in cells and was therefore predicted to lead to a gain of function [23]. Based on our analysis and by analogy with mutations *FGFR1* p.Cys178Ser and *FGFR2* p.His167_Asn173del, mutation *FGFR4* p.Cys172Gly is predicted to be an activating mutation that triggers the MEK pathway, similarly to *GNAQ* p.G48L.

### 2.2. Case Description

The patient, a 57-year-old female Caucasian, was diagnosed with uveal melanoma in the posterior and superior quadrants of the choroid of the right eye in September 2014. Following the eighth edition of the American Joint Committee on Cancer (AJCC), the tumor was classified as T4cN0M0 and therefore stage IIIB. The size of the tumor was 23.2 × 21.7 mm, with a thickness of 9.6 mm. There was no ciliary involvement and an extrascleral extension of 4.9 × 4.6 mm, and a thickness of 1 mm, was detected. The patient received a local therapy by proton beam radiotherapy. In June 2015, three liver metastases were detected by a control magnetic resonance imaging (MRI), and treated by local thermal-ablation. The patient progressed in May 2017 with lung, subcutaneous, and liver metastases. Systemic immunotherapy with the combination of ipilimumab and nivolumab was started. After three cycles, the patient experienced autoimmune thyroiditis, and the treatment was stopped. The thyroiditis resolved within a month. The patient then received one additional cycle of nivolumab, complicated by steroid-resistant autoimmune hepatitis, and the immunotherapy was definitely discontinued. In October 2017, the patient presented with the progression of subcutaneous nodular lesions, while lung and liver lesions remained stable. The liver lesions were again treated with thermal-ablation combined with hepatic radio-embolization. In April 2018, systemic progression and five new brain metastases were detected. Brain metastases were treated with stereotactic radiosurgery (SRS). Next, an in-house developed NGS, including the complete exons of 394 cancer-associated genes, was requested on one of the subcutaneous metastases to identify actionable genomic alterations. Three potentially pathological mutations were detected: *BAP1* c.68-4_84delinsGA (p.?), *FGFR4* c.514T > G (p.Cys172Gly) and *GNAQ* c.142_143delinsTT (p. Gly48Leu), with allelic frequencies of 82%, 47% and 41%, respectively (Appendix A). Based on the regions covered by our panel, we determined a relatively low tumor mutation burden (TMB) (2 non-synonymous somatic mutations/Mb), which is typical of uveal melanoma and in part could explain the absence of response to immune therapy [24,25]. Immunohistochemistry analysis showed a PD-L1-negative tumor (Tumor Proportion Score, TPS = 0%). The functional significance of the *GNAQ* mutation was described as uncertain in publicly available variant databases.

Consequently, molecular modelling was requested, which predicted a potential activating role of the *GNAQ* mutation. The *FGFR4* mutation was considered not targetable by specific FGFR1-3 inhibitors, such as erdafitinib or by non-specific kinase inhibitors such as sorafenib. In addition, we considered the *GNAQ* mutation downstream of FGFR activity and hence expected an effect of MEK inhibition also on FGFR4 signalling. Based on the results of NGS and molecular modelling, the MTB recommended MEK inhibitor therapy with trametinib at 2 mg/day, every day, which was started in May 2018. MEK inhibitors have long been tested in uveal melanoma due to the activation of the MAPK pathway by *GNAQ*. So far, only a limited efficacy of MEK inhibitors was detected, in immune therapy naive patients with classical *GNAQ* mutations [26,27,28]. Despite the absence of strong clinical evidence for MEK inhibitors in uveal melanoma and in the absence of other alternative therapies, we proposed trametinib. An additional reason for proposing a MEK inhibitor in our immune therapy exposed patient is that in patients with *NRAS* mutant melanoma, MEK inhibitors showed a better response rate and progression-free survival (PFS) in immune therapy-exposed patients than in immune therapy-naïve patients in the NRAS-mutant melanoma (NEMO) trial [29]. The reason for this apparent difference remains unclear. After two months of treatment, we detected a response (Figure 5). The patient experienced a grade III mucositis in August 2018, and the treatment had to be suspended for one month. During this time, we detected one new brain lesion, which was treated by Stereotactic Radiosurgery (SRS) (24Gy). In order to avoid further mucositis, the dose of trametinib was halved (1 mg/day). A repeat MRI showed two more new brain metastases, which, again, were successfully targeted with SRS (24Gy). After the initial response, the patient maintained a stable disease but eventually progressed after ten months of treatment, and trametinib was discontinued. Seventy-four months after the primary diagnosis and 32 months after the presentation of the case in the TBM, the patient remains alive.

## 3. Discussion

Using molecular modelling, we predicted that mutation *GNAQ* p.Gly48Leu in protein Gαq introduces new favorable hydrophobic interactions with SW-III that can maintain SW-III close to the active site and contributes to maintaining the active conformation of Gαq. On the contrary, this mutation is predicted to have little or no impact on the inactive conformation, as SW-III is far from the active site. Recently, Joseph N.M. et al. [18] hypothesized that mutation p.Gly48Leu could increase Gαq activity. We agree with their conclusions and established a possible mechanism for this activation. Our analysis strongly suggests that *GNAQ* p.Gly48Leu is a tumor-driver mutation possibly activating Gαq. In turn, Gαq can activate the MEK pathway via several mechanisms involving PLCβ and PKC or RasGEF. In addition, we also predicted that mutated *FGFR4* p.Cys172Gly likely leads to a gain of function by stabilizing the homodimer conformation, analogous with what was observed in *FGFR1* p.Cys178Ser and *FGFR2* p.His167_Asn173del [22,23]. Similarly to *GNAQ* p.Gly48Leu, this mutation-driven activation of FGFR4 could trigger the MEK pathway.

These conclusions contributed to ultimately select a MEK inhibitor, trametinib, as a personalized treatment for this patient. The fact that the patient had previously been treated with immune therapy also supported this decision, since, based on the results of the NEMO trial, immune therapy-experienced patients might respond better to MEK inhibitors and immune therapy than naive patients [29]. At the end of June 2019, after two months of treatment, the PET-CT (Positron emission tomography-computed tomography) showed a stable disease with a partial response of some subcutaneous nodules (Figure 5B). Although the analysis of the *GNAQ* p.Gly48Leu mutation predicted the activation of the protein function and consequently of the MEK pathway, strongly supporting the use of trametinib, we cannot exclude another mechanism by which trametinib might have contributed to stabilize the patient’s situation, including for instance a possible impact on the tumor microenvironment in melanoma, as mentioned by Kuske et al. [30].

This work illustrates the importance of molecular modelling analysis in personalized oncology. Depending on data and resources, it can lead—within the timeframe of the clinical evaluation of the case—to a valuable prediction of the possible impact of unknown mutations on the structure and activity of the modified protein, and by extension on the biological pathway involved, helping clinicians to propose the best treatment for their patients.

## 4. Materials and Methods

This case report includes a patient from Lausanne University Hospital, Switzerland and was conducted in accordance with the Declaration of Helsinki, the Swiss legal requirements and the principles of Good Clinical Practice. The protocol was approved by the Research Ethics Committee of Canton de Vaud, Switzerland (Protocol No. 2019-00448, 11 February 2020). The patient provided written informed consent to use her medical information for research purposes and report this case. The study and the patient treatment were approved by the MTB referents. Further information on research methods are available in the Appendix A.

For NGS analysis, sections of a formalin-fixed paraffin-embedded (FFPE) biopsy of a subcutaneous metastasis, containing approximately 80% of tumor cells, were collected for DNA extraction (Maxwell 16 FFPE Plus LEV DNA Purification kit, Promega, Madison WI, USA). Matched constitutional DNA was extracted from blood (Maxwell 16 LEV Blood DNA kit, Promega). Starting from 100 ng DNA, capture-based targeted high-throughput sequencing was performed with a KAPA HyperPlus library preparation kit (Roche, Pleasanton, CA, USA), followed by hybridization capture using a custom design of xGen Lockdown Probes (Integrated DNA Technologies, Coralville, IA, USA) covering the full-coding sequences of 394 cancer-associated genes (full list available upon request). Enriched libraries were sequenced using an MiSeq instrument (Illumina, San Diego, CA, USA). Sequence analysis was based on established algorithms and pipelines according to the Genome Analysis Toolkit (GATK) standards. Briefly, forward and reverse reads were aligned to the human genome (GATK repository, build 37 decoy) using BWA-MEM (v0.7.5a). Binary alignment map (BAM) files were subjected to PCR duplicate removal (Picard v2.1.0), followed by realignment around indels and base recalibration using GATK tools (v3.7). Single nucleotide variant (SNV) and indel variant calling was performed using samtools mpileup (v1.9-2) and VarScan (v2.4.3) as well as the MuTect2 algorithm (GATK v3.7) comparing tumor versus matched normal samples. Raw variant calls were annotated for presence in the dbSNP and COSMIC databases as well as the mutation effect on gene transcript by SnpEff (v.4.3). SNVs and indels were filtered based on coverage, quality and variant allele frequency (threshold 5%). Furthermore, variants were filtered using their biological impact as well as a panel-specific list of known artefacts, which were collected during the validation phase of the panel. All retained alterations were confirmed by visual inspection with IGV software.

UCSF-Chimera (v1.13.1) was used to analyze structures and to measure atom distances [31]. Folding free energies were calculated with FoldX4, which is freely available for academic and non-profit research institutions [19], with the PositionScan option. Amino acid sequences were retrieved from UniProt [32,33] and the multiple sequence alignment was performed with Muscle [34]. Gαq structures were retrieved from the Protein Data Bank (PDB) [35]: 3ah8 [8] (sole inactive conformation), 2bcj [36], 2rgn [36], 3ohm [7], 4gnk [37], 4qj3 [38], 4qj4 [38], 4qj5 [38], 5do9 [39]. The FGFR4 experimental 3D structure used for the analysis was retrieved from the PDB: 1qct [21].

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
