# Peer review of "Trametinib Induces the Stabilization of a Dual GNAQ p.Gly48Leu- and FGFR4 p.Cys172Gly-Mutated Uveal Melanoma. The Role of Molecular Modelling in Personalized Oncology"

_ijms, 2020, doi:10.3390/ijms21218021_

Round 1
Reviewer 1 Report
This manuscript documents the molecular modeling of p.Gly48Leu in GNAQ and p.Cys172Gly in FGFR4 in a subcutaneous metastasis of a uveal melanoma primarily treated with proton beam irradiation. In GNAQ, the authors suggest an increased stability of the active form of GNAQ through hydrophobic interactions of Gly48Leu with Leu249 and Val240 on the Switch III loop. They used foldX to assess the gain of energy of each conformation with and without the mutation. In FGFR4, p.Cys172Gly is assumed to stabilize FGRFR4 as already described for FGFR1 p.Cys178Ser and FGFR2 p.His167_Asn173del. Assuming that both these mutations could activate the MAP kinase pathway in the metastases, the patient was treated with Trametinib with a partial response over a 10 months period.
This report is highly interesting and demonstrates that next generation sequencing combined with molecular modeling can efficiently improve the patient’s outcome.
The response of this case and the molecular modeling suggest an oncogenic effect of pGly48Leu, but the definite proof of this effect would likely require substantial more work with GNAQ constructs with and without the mutation that would go beyond the scope of this report. Were there any short term cultures derived from the metastasis to directly test Trametinib efficiency in vitro? If not, how was MAP kinase activity in the subcutaneous metastases?
P7 line 186, what were the size and the localization (ciliary body, choroidal or both) of the primary uveal melanoma in 2014?
P8 line 227, what is the total follow up?
Author Response
Lausanne, October 19th, 2020
Re: ijms – 957942: Revision of manuscript “Trametinib induces the stabilization of a dual GNAQ p.Gly48Leu and FGFR4 p.Cys172Gly-mutated uveal melanoma. The role of Molecular Modelling in Personalized Oncology” by Fanny .S Krebs, Camille Gérard, Alexandre Wicky, Veronica Aedo-Lopez, Edoardo Missiaglia, Bettina Bisig, Mounir Trimech, Olivier Michielin, Krisztian Homicsko, Vincent Zoete.
Dear Reviewer,
We would like to thank you for their careful reading of our manuscript and for identifying relevant points. We hope our corrections meet your expectation.
Please, find bellow our answer, and please see attachment for a word version.
Answers
Were there any short term cultures derived from the metastasis to directly test Trametinib efficiency in vitro? If not, how was MAP kinase activity in the subcutaneous metastases?
Unfortunately, no short term cultures derived from the metastasis was performed. The MAP kinase activity in the subcutaneous metastases was not tested.
P7 line 186, what were the size and the localization (ciliary body, choroidal or both) of the primary uveal melanoma in 2014?
The primary uveal melanoma diagnosed in 2014 was of the choroid of the right eye, in both posterior and superior quadrants. The size of the tumor was 23.2 mm x 21.7 mm, with a thickness of 9.6mm. There was no ciliary involvement but we detected the presence of an extrascleral extension of 4.9 mm x 4.6 mm, with a thickness of 1 mm. This information was added in the manuscript (line 217-220)
P8 line 227, what is the total follow up?
74 months after the primary diagnosis, the patients remains alive. This information was added in the manuscript (line 264-265)
Reviewer 2 Report
The authors describe the finding of two variants of unknown significance by NGS in a patient with uveal melanoma. By investigating the 3D protein structure of wild type and mutated GNAQ and FGFR4 they find evidence of these mutations being activating. Treatment of the patient with the MEK inhibitor trametinib resulted in initial response and stable disease for 10 months.
The study nicely shows the benefit of molecular analysis and discussions in molecular tumorboards for patients without any treatment options left. As extensive molecular analysis and molecular tumorboards are more and more performed, reports of mutations with yet unknown clinical significance that may offer additional treatment options after investigating the mutation by molecular modelling are extremely important.
Detailed requests:
- Supplementary figures are not in the correct order. In the text it starts with A2, then A5…
- Same thing for the suppl. tables
- Make sure all gene names are written in italics
- Line 43: should be “in this tumor type”
- Line 54: FGFR members
- Line 55: …present in melanoma cancer -> delete “cancer”
- Line 57/58: … of a patient known to have uveal melanoma
- Line 60/86ff: Although there is no literature about functional data, please mention the publication that has reported this mutation before in uveal melanoma, Johansson et al. Oncotarget 2016
- Line 179: should be figure 4
- Line 229: should be figure 5
- Figure 5 and corresponding text: Please indicate SUVmax values and the difference in size for the lesions (especially the responding lesions). Is there a PET scan between 2 and 10 months available to show stable disease?
- Line 251: This should be figure 5
- It would be interesting to see whether the patient acquired any resistance mutations after progression under trametinib. Is tumor material available for NGS?
- Line 304ff and 311ff: I guess this can be deleted (?)
Author Response
Lausanne, October 19th, 2020
Re: ijms – 957942: Revision of manuscript “Trametinib induces the stabilization of a dual GNAQ p.Gly48Leu and FGFR4 p.Cys172Gly-mutated uveal melanoma. The role of Molecular Modelling in Personalized Oncology” by Fanny .S Krebs, Camille Gérard, Alexandre Wicky, Veronica Aedo-Lopez, Edoardo Missiaglia, Bettina Bisig, Mounir Trimech, Olivier Michielin, Krisztian Homicsko, Vincent Zoete.
Dear Reviewer,
We would like to thank you for their careful reading of our manuscript and for identifying relevant points. We hope our corrections meet your expectation.
Please, find bellow our answer, and please see attachment a word version.
Answers
- Supplementary figures are not in the correct order. In the text it starts with A2, then A5… Same thing for the suppl. Tables
- Thank you for noticing this problem. The supplementary figure and table orders was corrected to follow the chronological order in the manuscript (line 35, 45, 47, 76, 118, 124, 125,132, 133, 167) and in the supplementary file.
- Make sure all gene names are written in italics
- All gene names and are now written in italics in the manuscript (line 2, 55, 69, 181, 190, 193, 195, 196, 197, 205, 207, 212, 239, 245, 247, 251, 252, 258, 296) and in the supplementary file.
- Line 43: should be “in this tumor type”
- The sentence at line 44 was corrected from “in this tumor” to “ in this tumor type”.
- Line 54: FGFR members
- FGFRs members” was corrected to “FGFR members” (line 55)
- Line 55: …present in melanoma cancer -> delete “cancer”
- The word “cancer” was deleted at line 56
- Line 57/58: … of a patient known to have uveal melanoma
- The sentence “subcutaneous metastasis of a patient known for an uveal melanoma” was corrected to “subcutaneous metastasis of a patient known to have an uveal melanoma” (line 58)
- Line 60/86ff: Although there is no literature about functional data, please mention the publication that has reported this mutation before in uveal melanoma, Johansson et al. Oncotarget 2016
- The publication of Johansson et al. Oncotarget 2016, which reported the mutation, cited by the reviewer was included in the manuscript (line 63). All the following reference numbers were changed according to this addition.
- Line 179: should be figure 4; Line 229, line 251: should be figure 5
- The figure numbering for Figure 5 was corrected (line 297). The figure 4 is correctly attributed.
- Figure 5 and corresponding text: Please indicate SUVmax values and the difference in size for the lesions (especially the responding lesions). Is there a PET scan between 2 and 10 months available to show stable disease?
- The sizes and SUVmax values of the lesions at the different stages were added to the description of Figure 5 (line 275-277). Unfortunately, no PET scan between 2 and 10 months is available.
- It would be interesting to see whether the patient acquired any resistance mutations after progression under trametinib. Is tumor material available for NGS?
- There is post-resistance tumor material available but no new NGS analysis was done. It would indeed be interesting to analyze our list of 400 genes to look for new potential mutations potentially involved in trametinib resistance. However, it would require more than one month to obtain the results, which is beyond the deadline to submit the revision.
- Line 304ff and 311ff: I guess this can be deleted (?)
- The Appendix description, between the abbreviations and references section, was deleted (line 367).